# Serum Concentrations of Selected Organochlorines in Pregnant Women and Associations with Pregnancy Outcomes. A Cross-Sectional Study from Two Rural Settings in Cambodia

**DOI:** 10.3390/ijerph17207652

**Published:** 2020-10-20

**Authors:** Margit Steinholt, Shanshan Xu, Sam Ol Ha, Duong Trong Phi, Maria Lisa Odland, Jon Øyvind Odland

**Affiliations:** 1Department of Public Health and Nursing, Norwegian University of Science and Technology, 7491 Trondheim, Norway; shanshan.xu@ntnu.no (S.X.); jon.o.odland@ntnu.no (J.Ø.O.); 2Helgelandssykehuset, 8801 Sandnessjoen, Norway; 3Trauma Care Foundation, Battambang, Cambodia; olliekhm@gmail.com; 4Department of Environment and School Health, Nha Trang Pasteur Institutte, Nha Trang, Khánh Hòa 650000, Vietnam; dtrongphi@yahoo.com; 5Institute of Applied Health Research, University of Birmingham, Birmingham B152TT, UK; marialisaodland@gmail.com; 6Department of General Hygiene I.M. Sechenov First Moscow State Medical University (Sechenov University), Trubetskaya str., 8-2, 119992 Moscow, Russia

**Keywords:** polychlorinated biphenyls, organochlorine pesticides, low-resource settings, pregnancy outcomes, neonatal health, stunting

## Abstract

We conducted a cross-sectional study among 194 pregnant women from two low-income settings in Cambodia. The inclusion period lasted from October 2015 through December 2017. Maternal serum samples were analyzed for persistent organic pollutants (POPs). The aim was to study potential effects on birth outcomes. We found low levels of polychlorinated biphenyls (PCBs) and organochlorine pesticides (OCP), except for heptachlors, β-hexachlorocyclohexane (HCH), heptachlor epoxide, and p,p’-DDE. There were few differences between the two study locations. However, the women from the poorest areas had significantly higher concentrations of p,p’-DDE (*p* < 0.001) and hexachlorobenzene (HCB) (*p* = 0.002). The maternal factors associated with exposure were parity, age, residential area, and educational level. Despite low maternal levels of polychlorinated biphenyls, we found significant negative associations between the PCB congeners 99 (95% CI: −2.51 to −0.07), 138 (95% CI: −1.28 to −0.32), and 153 (95% CI: −1.06 to −0.05) and gestational age. Further, there were significant negative associations between gestational age, birth length, and maternal levels of o,p’-DDE. Moreover, o,p’-DDD had positive associations with birth weight, and both p,p’-DDD and o,p’-DDE were positively associated with the baby’s ponderal index. The poorest population had higher exposure and less favorable outcomes.

## 1. Introduction

The European Union defines pollution as the introduction of unwanted, mostly dangerous, material into the Earth’s environment as the result of human activity, and which poses a threat to human health and harms the ecosystems [1]. Since 1950 more than 140,000 new chemicals and pesticides have been produced and disseminated worldwide at unknown measures; 5000 of these substances are manufactured at such volumes that they can be found almost anywhere on the globe [2]. Less than half have been tested for adverse effects on human health, and only a few high-income countries perform premarketing evaluation before releasing new products [2].

Although international conventions aim to mitigate and reduce the production and distribution of some pollutants [3], new chemicals enter air, soil, and water and thus continue to expose humans to health hazards. In addition, production and distribution of these potentially harmful substances are, in this day and age, often moved to low-cost countries with less rigorous and transparent legislations [2].

Only a small proportion of chemicals are tested for neurotoxic effects in adults. Even fewer are scrutinized for adverse effects in the developing brains of fetuses and growing children [4]. Nevertheless, the impact of persistent toxic substances (PTS) on fetuses and small children has been well documented for decades [5]. Both synthetic substances such as organochlorines (OCs) and naturally occurring non-essential toxic trace elements are classified as PTS due to their very long half-lives under most environmental conditions and the fact they may be toxic even in very small amounts. In addition, the substances both bioaccumulate, as an individual grows older, and bio magnify, with increasing concentrations in the food chain. The long-term impact of exposure to these pollutants is not fully understood. Of particular interest is the “cocktail effect” on the developing neural systems in fetuses and young children since organochlorines are lipophilic and thus accumulate in fatty tissue [6,7,8].

Cambodia is a small low-income country in Southeast Asia with a population of 16.2 million. It is ranked 146th out of 189 countries in human health indicators. According to the United Nations Development Program (UNDP), close to 40% of Cambodia’s population live in so-called multidimensional poverty, and 32.4% of all children <5 years of age suffer from moderate or severe malnutrition or stunting [9,10].

Vulnerable populations are more exposed to pollutants both ambient and through the food chain. In this study, we will look at levels of selected persistent organic pollutants (POPs) (organochlorine pesticides (OCPs) and polychlorinated biphenyls (PCBs)) in the blood of pregnant women from two rural communities in Cambodia and their effect on birth outcomes such as gestational age, length, weight, ponderal index, and head circumference of their babies. We will also discuss possible sources of exposure from environmental surroundings and diet.

## 2. Materials and Methods

### 2.1. Study Sites and Population

Our cross-sectional study was conducted from October 2015 through December 2017. The inclusion period for the women lasted from October 2015 through May 2016. The inclusion criteria are described in our first article from the study [11]. Two areas in Northwestern Cambodia were selected; both considered low-income non-urban communities. Area one, Chroy Sdao district, is situated approximately 30 km north of the provincial center, the city of Battambang. The population of 21,000 live in 11 villages that have been involved in rice farming for generations. Area two is located in Eak Phnom district; also called “the floating villages” since the majority of the population live a semi-nomadic life onboard boats and rafts on the waterways between the cities of Battambang and Siem Reap. More than 90% of the 18,000 inhabitants earn their livelihoods from the fisheries [11].

A total of 194 women, 120 from area one and 74 from area two, were recruited while in the third trimester of pregnancy. Originally, we aimed for two study groups more similar in size. However, due to difficult access to the floating villages, it turned out to be challenging to recruit more women from area two. The villages are close to inaccessible during the dry season, and in the rainy season many of the families go far to reach the best fishing grounds. We collected blood and urine samples together with socioeconomic, anthropometric, and medical data. Data about the newborns as well as delivery information from their mothers were recorded by health personnel at time of birth.

There were 172 babies included in the study: 81 boys (47%) and 91 girls (53%). We missed data from 22 children. The main reason for this was the semi-nomadic life of the families, especially for the ones living on the waterways. They follow the seasons and move their rafts and homes to wherever the fishing is better. Their babies were most probably born at home, and thus no reliable data were registered at the health center. For the inland women, some migrate to Thailand during harvesting seasons, and they may have delivered their babies abroad.

### 2.2. Ethical Approval

The participant’s consent was given both orally and in writing. The informed consent forms were presented to every woman by local health staff to ensure full understanding of the study aims and process. The document was signed or thumb printed on site when the blood samples were drawn. The consent forms and written files of demographical and laboratory data were stored in locked room at the Trauma Care Foundation head office in Battambang, Cambodia. Access to non-anonymous data was restricted to members of the research team. The data were stored and processed according to the approval in National Ethics Committee for Health Research of the Ministry of Health, Cambodia (ref. 0365 N.E.C.H.R., 29/12/2014 and 114 N.E.C.H.R; 28/03/2016), as well as the Regional Committee for Medical Research Ethics, 2016, ref: 2015/2486/ REK Nord, Norway.

### 2.3. Questionnaire

The women were examined and interviewed by the presiding midwife and local researchers. Illiterate women were assisted by the research assistants. The questionnaire was similar to those used in comparable studies of the Arctic Monitoring and Assessment Programme (AMAP). It was translated into Khmer and adapted to and with additional questions adjusted to Cambodian context. The gathered information included name, age, height, and weight at the time of inclusion, reproductive history, and socioeconomic background. The questionnaire also documented lifestyle, environmental history, and consumption frequencies of the most commonly eaten food. Appendix A has further information about the questionnaire.

### 2.4. Chemical Analysis and Quality Control

Non-fasting venous blood was drawn from each woman at the time of inclusion; 14 mL divided into two separate sterile BD vacutainer^®^ plastic tubes, one containing EDTA and the other containing heparin as anticoagulant. EDTA is the preferred anticoagulant for heavy metal analyzes, while the full blood in the heparin tube was intended for the OCP and PCB analyses. Each tube was gently inversed 5−6 times before placed in a cooling box at temperature of four degree Celsius (≤4.0 °C). Within 12 h, the blood in the EDTA tube was centrifuged at 2000 rpm for 15 min. The plasma was then transferred to a 12 mL (16 × 100 mm) cultured glass tube with cap. The whole blood sample from the heparin tube was transferred to a 12 mL cultured glass tube. All samples were stored in a freezer with temperature at minus 20 degree Celsius (−20 °C) until they were referred to the Pasteur Institute. The freezer was securely placed in a locked room at the Trauma Care Foundation head office in Battambang, Cambodia.

The plasma samples were transported to Pasteur Institute in Nha Trang, Vietnam, by TSP Express Cambodia Co. Ltd. on 28 October 2016. The whole blood samples, frozen and stored in cooling box, were taken to Pasteur Institute by car from Battambang by the research assistant.

The samples were analyzed at Nha Trang Pasteur Institute, Vietnam, for 18 PCBs and 19 organochlorine pesticides as well as the most relevant toxic and essential metals.

The chemical analysis was performed according to the international quality control system QA/QC established by the Centre de Toxicologie du Quebec (https://www.inspq.qc.ca/CTQ). Plasma samples were extracted by liquid–liquid extraction with ethanol, de-ionized water saturated with ammonium sulfate, and hexane. The OCs were separated from the lipids by a Florisil column manually packed with 3.0 g of 0.5% deactivated Florisil and 2 g of granulated sodium sulfate on top of the columns. The OCs were eluted with 11 mL hexane: dichloromethane (3:1 *v*/*v*). The collected fraction was evaporated to 0.5 mL by Centrivap Concentrator. The sample was evaporated to 200 μL before transfer to a gas chromatography (GC)-vial with an insert capillary, then reduced further to 20 μL, and octa-chloronaphthalene (OCN) was added as a recovery standard.

The gas chromatography (GC) was performed using a Shimadzu GC 2010 fitted with a Shimadzu AOC 20 S auto sampler (Injector: AOC 20 I) connected to a Shimadzu MS QP2010 plus spectrometer. A 30 m SLB−5 MS column (0.25 mm i.d. and 0.25 μm film thickness; Supelco) was used. The instrument was operated in the selected ion monitoring (SIM) mode, employing positive electron-impact ionization (EI+) as the source for PCB and OCPs.

Next, 13 C-labeled PCB 77 was used as an internal standard for PCB 28/31, PCB 33, PCB 52, PCB 74, PCB 95, PCB 101, and PCB 99. Similarly, 13 C-labeled PCB 118 was used as an internal standard for PCB 149, PCB 118, PCB 153, PCB 105, PCB 138, and PCB 158, as well as an internal standard for the rest of the PCBs. Finally, 13 C-labeled HCB was used as an internal standard for HCB, 13 C-labeled β –HCH for HCH, 13 C-labeled p,p’-DDT for DDT, and 13 C- labeled p,p’-DDE for the rest pesticides. The different compounds were identified from their SIM masses and retention times. For each analyte, the ratio of two masses was monitored. Peaks with differences in isotopic ratio greater than 20% compared to the quantification standard were rejected and not quantified. The samples were analyzed for 22 PCBs and 19 pesticides. For every 10 samples, a blank was analyzed to assess laboratory-derived (i.e., inadvertent) sample contamination. The method detection limit (LOD) was calculated as: LOD = Mean of blank concentration + 3 × SD of blank concentration.

### 2.5. Measurement of Birth Outcomes

Outcomes variables were gestational age (weeks), birth weight (kilograms), birth length (cm), head circumference (cm), and ponderal index (kg/m^3^) retrieved from the medical records. Ponderal index was used to estimate the nutritional status for the newborns, ponderal index = weight(kg)/height (m^3^). We did not collect any information about the placenta.

### 2.6. Statistical Analysis

Statistical analyses were carried out using IBM SPSS Statistics for Windows (version 26; SPSS Inc., Chicago, IL, USA).

Sociodemographic data are presented as arithmetic means, standard deviations (SDs), median, minimum and maximum, or proportion (%). Due to the non-normal distribution of data, Mann–Whitney U test was applied to compare the socioeconomic and demographic differences between the two sites. Geometric means (GMs), 95% confidence interval (CI), median, minimum, and maximum were used for the POPs’ descriptive analyses. Despite log10 transformation, the blood PCBs and OCPs concentrations remained skewed and were not normally distributed according to the Shapiro–Wilk test and Q–Q plot. Statistical differences in the levels of POPs between the two study areas were tested for significance using Mann–Whitney U test. Multivariate association was evaluated with multiple linear regression models to describe the relationship between selected POP exposure and related maternal risk factors. The effects of maternal PCBs and OCPs levels on birth outcomes were assessed by multiple linear regression models. The potential covariates and confounders adjusted for in the regression models were selected based on previous literature [7,12] or on their association with maternal blood POPs and or birth outcomes (*p* < 0.05) in this study. Covariates included maternal age (years), parity (para 0, para (1−3), and para (≥4)), body mass index (BMI, kg/m^2^), residence area (inland/floating area), education (years), occupation (housewife, worker, farmer, teacher, and fishing). In addition, ponderal index was added into the gestational age regression model, and gestational age was introduced into birth weight, birth length, head circumference, and ponderal index regression models [7].

Complete case analysis was used meaning participants with any missing data were excluded in the statistical analyses. Before inclusion in analysis, maternal blood POP concentrations were log10 transformed. A significance level of *p* < 0.05 (two tailed) was used for all analyses.

## 3. Results

### 3.1. Demographic, Socioeconomic, and Anthropometric Findings

The demographic, socioeconomic, maternal, and neonatal clinical characteristics are presented in Table 1. The participants were from 17 to 44 years old, with a mean age of 26.7 ± 6.2 years. There was no difference in maternal body mass index between the areas; however, the women from area two, were significantly shorter and weighed less at the time of inclusion than the women from area one. The educational length was strongly linked to geographical locations, with an average of 7.1 educational years in the inland area compared to 4.6 years among women from the floating villages. Mean gestational age was 39.1 ± 1.0 weeks, ranging from 38 to 42 weeks in both locations. The gestational age was significantly longer in study area one: 40.0 versus 38.4 weeks. A large proportion of the inland women (45%) were farmers, while in the floating area the majority (52.7%) had fishing as their livelihood. Almost all women (93%) from the floating villages reported that the river or ponds were the family’s main source of drinking water, while the majority of the inland women collected rainwater (53.3%) or had access to wells (37.5%). In the floating areas, close to 80% of the households used insecticides; the majority only spraying their living quarters. In comparison, less than half of the families (42%) in the inland areas used insecticides, and among them approximately 70% used insecticides both at home and on the farmland. All women from the inland area had access to rice, meat, fruit, and vegetables from their own village. They either grew themselves or bought them from the local market. In contrast, most women (98%) from the floating area had to use a boat to reach a market where they could purchase staple food items.

The average birth weight, length, and head circumference were 3.1 kg, 48.9 cm, and 31.7 cm, respectively. Although arithmetic average birth weight was similar in inland and floating area (3.2 vs. 3.0 kg), neonatal birth weight from the inland area had a larger mean rank (85.1) than those in floating area (with mean rank 62.3) (Data not shown), a statistically significant different was observed (U = 1901.0, *p* = 0.001). The babies from the floating area had a significantly lower ponderal index and smaller head circumference than the neonates from inland (Table 1).

### 3.2. Distribution of Selected Polychlorinated Biphenyls and Organochlorine Pesticides

Descriptive statistics of selected POP concentrations in maternal blood are summarized in Table 2. The level of each pollutant is presented as wet-weight; concentration of the organochlorine/unit serum.

The most abundant OCP was p,p’-DDE, with a median of 0.720 picogram/microliter, followed by heptachlors, β-HCH, and heptachlor epoxide (0.135, 0.103, and 0.069, respectively). p,p’-DDE made up almost all of the DDT metabolites (90.2%), while β-HCH was the most abundant isomer contributing 44.2% of total HCH. PCB 180 was the most common PCB congener (24.8%) followed by PCB 153 (17.9%), PCB 118 (15.4%), and PCB 138 (12.8%).

Most maternal PCB and OCP sera concentrations were similar between inland and floating areas (Figure 1). The levels of PCB 52, α-HCH, β-HCH, γ-HCH, heptachlor epoxide, and o,p’-DDT remained constant in all samples, thus these compounds were excluded for further statistical analyses. Statistical differences between the residential areas were observed for HCB (*P* < 0.001) and p,p’-DDE (*P* = 0.002). HCB and p,p’-DDE levels were all higher in women from the floating area compared to those in women from the inland site. Appendix A presents the exact concentrations of the blood POP levels in each area.

### 3.3. Association between POPs Serum Concentrations and Maternal Characteristics

Maternal age, BMI, parity, education, residence area, insecticide use, occupation, food, and water source were factors potentially associated with POP exposure. Multiple regression models of these factors and POP concentrations provided a relatively comprehensive description of the main factors related to the levels of these pollutants (Table 3). Parity was the main factors for PCB 138 (*p* < 0.05), PCB 153 (*p* < 0.05), HCB (*p* < 0.05), and p,p’-DDE (*p* < 0.001) showing a negative significant correlation, i.e., multiparous women had significantly lower levels of these compounds compared to women expecting their first child. However, women who had more than four previous pregnancies had significantly higher levels of mirex compared to primiparas (*p* < 0.001). Aside from parity, maternal age was a main determinant for PCB 153 and p,p’-DDE levels, presenting a positive association with PCB 153 (*p* < 0.05) and p,p’-DDE (*p* < 0.001). Educational level was also important; women with more years in school had statistically higher levels of p,p’-DDE and p,p’-DDT (*p* < 0.05 for both). Women who used insecticides had a higher level of p,p’-DDE than non-users. The PCB 138 concentration was significantly higher among women buying food from the market compared to women who could access groceries in their village. Adjusting for occupational factor, it is interesting to note that women who work in the fishing industry have significantly higher levels of mirex compared to housewives. The concentrations of PCBs 118 and 153 were higher in teachers compared to those informants defining themselves as housewives. We found no significant associations between PCB and OCP levels and the different sources for drinking water.

### 3.4. Maternal Concentrations of Persistent Organic Pollutants and Pregnancy Outcomes

Multivariable analysis between exposure to the selected persistent organic pollutants and pregnancy outcomes are presented in Table 4. The main finding is the significant association of o,p’-DDE exposure with gestational age, birth length, and ponderal index (*p* < 0.05 in all cases). There were significant negative associations between maternal blood levels of PCB congeners 99, 138, and 153, and o,p’-DDE and gestational age. The association between gestational age and PCB 180 levels remained marginally and inversely significant (*β* = −0.15; 95% CI: −3.16 to 0.02; *p* = 0.052). No associations were observed for POP exposure and birth weight, except for o,p’-DDD, which was positively associated with birth weight (*β* = 0.21; 95% CI: 0.08 to 3.64; *p* = 0.041). PCB congeners 118 and 180, o,p’-DDE and p,p’-DDD showed significant negative associations with birth length, while aldrin exposure indicated a positive relation with birth length. The levels of o,p’-DDE and p,p’-DDD were positively associated with ponderal index, while aldrin showed the opposite outcome. No statistically significant associations were found for the selected OCPs and PCBs in this study and the babies’ head circumference.

## 4. Discussion

In our study, we found low levels of polychlorinated biphenyls (PCB) and o,p’-DDE, o,p-DDD, p,p’-DDD, mirex, and aldrin except for heptachlors, β-HCH, heptachlor epoxide, and p,p’-DDE. The results from the two study locations were quite similar; however, the women from the floating areas had significantly higher concentrations of p,p’-DDE and HCB in the blood compared to women situated in the inland villages. Parity was the main factor associated with POP exposure, with multiparous women having significantly lower levels of PCB 138 and 153, HCB and p,p’-DDE compared to primiparas. Maternal age was positively associated with PCB 153 and p,p’-DDE concentrations. There was positive association for p,p’-DDE and women who sprayed their homes and/or farmland with insecticides. Educational level was strongly associated with residency, with the women from the floating areas having less formal schooling. Educational length was positively associated with p,p’-DDE and p,p’-DDT. The PCB 138 level was significantly higher among women buying food from a market compared to that among women who accessed food in their own villages. This reflects the difference between the two sites, since almost all the women in inland area consumed locally grown produce, while the majority of women in the floating area had no other option than purchasing groceries from a market. Despite low maternal concentration of polychlorinated biphenyls in our material, we found a significant negative association between the PCB congeners 99, 138, and 153 and gestational age. Further, there were significant negative associations between gestational age, birth length, and maternal blood levels of o,p’-DDE. o,p’-DDD had positive associations with birth weight, and both p,p’-DDD and o,p’-DDE were positively associated with the baby’s ponderal index.

The finding of low levels of organic pollutants is in accordance with similar studies from South Vietnam [8] and Argentina [13]. The same goes for the relatively high concentrations of DDT metabolites indicating both historic and recent use of the chemical [8,13]. In contrast, Veyhe et al. found very low levels of all contaminants, including DDE, in a study from the Arctic part of Norway [14], with Norwegian women having less than 40% the levels of DDE metabolites compared to our Cambodian informants.

p,p’-DDE is the most common and abundant OCP detected both in the global environment and in human tissue. It is the main metabolite of p,p’-DDT, has a longer half-life, and is more toxic than the original chemical. In humans, p,p’-DDT is metabolized to p,p’-DDE within six months. The ratio between p,p’-DDE and p,p’-DDT thus provides useful information about an individual’s history of exposure of DDT, with a ratio <5 indicating recent use of the insecticide. The longer the time since exposure, the higher the numerator will be. A ratio >30 is therefore a strong indication for a dietary source of the chemical [8,15]. The ratio found in our study populations shows their main exposure to DDT is through the food chain. There is, however, a distinct difference between the two areas, and the women from the floating villages have the highest concentration of p,p’-DDE and, thus, also the highest ratio. Fresh water fish, a staple food in the floating villages, most probably add to the burden of exposure [11,16]. In addition, the population on the river and lake used more insecticides in their homes.

In our material, we found significant negative associations between gestational age, birth length, and maternal blood levels of o,p’-DDE. o,p’-DDE is the second most common DDT-metabolite and is associated with effects on hormonal receptors [17,18]. For o,p’-DDD we found positive associations with birth weight, and both metabolites were positively associated with the baby’s ponderal index. This is partly in accordance with Bravo’s study from 2017, where higher levels of 4,4 DDT (4,4 DDE composed >90% of the DDT metabolites in the study [13]; the same proportion as in our material) were found to increase gestational age as well as birth weight and length. Bravo’s findings are supported by laboratory studies showing that DDT exposure increased adipogenesis in vitro [19] and obesity in rats [20,21].

A possible explanation for o,p’-DDE being both negatively associated with gestational age and birth length and positively associated with ponderal index (PI) in our study could be that the metabolite induces symmetrical intrauterine growth restriction; thus normal PI [22].

We found very low concentrations of polychlorinated biphenyls in maternal blood from both study communities. This is in accordance with findings from Vietnam [8], and it supports the fact that there are few new sources for these pollutants in the Southern hemisphere [12,15]. Despite the low blood values, there were significant negative associations between the PCB congeners 99, 138, and 153 and gestational age. Bravo et al. found a negative association between HCB and PCB 153 and head circumference in neonates from Siberia [7]. In our material, we did not detect any association, either positive or negative, concerning levels of POPs and head circumference.

In a study from Bretagne, France in 2010, Petit et al. found smaller head circumference in babies born to mothers living in pea farming communities [23]. Pea farming in this actual study was associated with extensive use of organophosphate insecticides compared to growing other crops such as corn and wheat. Smaller head circumference was also associated with low educational level for the mother. The latter is in accordance with our own data from Cambodia [11]. A recent study from South Africa did not detect any association between maternal levels of DDT metabolites and neurodevelopment in their children at the ages of 1 and 2 years [24]. In Eskenazi’s material, the babies had an average birth weight similar to that in our study population, however, head circumference was not one of the study outcomes [24].

The primiparas in both locations had higher levels of PCB 138, PCB 153, HCB, and p,p’-DDE than women who had one or more previous pregnancies. These findings are similar to data from Vietnam [8] and also from the circumpolar studies conducted by AMAP [6]. POPs are lipophilic, accumulate in fatty tissue, cross the placenta, and are secreted into breast milk. Pregnancy and lactation are therefore important biological processes when the mother depletes her body burden of organic pollutants at the expense of the fetus and suckling baby [6,12,13].

Our study shows significant differences in levels of contaminants depending on residency and socioeconomic status, with the poorer women being more exposed. Their children are also the smallest with a significantly lower ponderal index and smaller head circumference. In the first article from our study, we presented the anthropometric findings supporting the hypothesis that a large proportion of our study population may be stunted [11]. Similar findings are reported by the UNDP showing that one third of Cambodia’s children <5 years of age are either stunted or suffer from malnutrition [9]. Maternal stature is connected to child survival [25,26], and since the poorer women in the study probably are stunted themselves and more exposed to persistent organic pollutants, their offspring are at higher risk of adverse health outcomes. Even though we did not find any significant association between maternal POP concentrations and head circumference, the fact remains that most of the babies from our study have alarmingly small head circumferences despite being safely above the WHO’s definition of low-birth-weight babies (LBW) >2500 g [27].

The developing brain and nervous system in fetuses and young children are extremely vulnerable. If children start their lives with depleted physical resources due to insufficient intrauterine nutrition, they may be more susceptible than a well-nourished baby to negative impact from lipophilic pollutants. In addition, living in poor socioeconomic settings is a risk factor in itself for adverse development of the child’s brain [28]. Chronic stress, such as that caused by insecure financial situations in the family, affects the hippocampus, amygdala, and areas of the prefrontal cortex negatively [29]. Both our study groups live in rural low-income settings, however, women from the floating villages are the poorest, the shortest, and have the lowest educational level. Their babies have lower ponderal index and smaller head circumference and are more exposed to pollutants in utero and will probably grow up in a less intellectually stimulating home environment. These are all factors known to negatively impact children’s cognitive development [30,31,32,33].

The number of participants may be argued to be low, and the two study groups are different in size. However, the findings are significant, and the study highlights the living conditions for rural populations rarely studied. To our knowledge, this is the first study of its kind from Cambodia combining anthropometric, socioeconomic, and pregnancy outcomes with analysis of maternal concentrations of selected organochlorinated pesticides. We have found babies with high risk of adverse physical and mental health development, and this calls for further investigations and also interventions to improve their living conditions.

## 5. Conclusions

In our study, from two low income settings in rural Cambodia we found low levels of persistent organic pollutants. The exception was pesticides, with DDT and its metabolites being the most abundant. The ratio between DDT and the most common metabolite p,p’-DDE indicates that diet is the main source of exposure. However, recent use of the pesticide is suspected. Nulliparous women had higher levels of contaminants than multiparous women. There was positive association for some DDT metabolites and ponderal index for the baby. This is puzzling; however, it may be due to symmetrical intrauterine growth retardation caused by insufficient nutrition for the pregnant women. The women from the poorest settings had higher blood concentrations of pollutants, and this is linked to negative pregnancy outcomes such as shorter gestational age and birth length. This is alarming and calls for interventions to protect populations at risk.

## Figures and Tables

**Figure 1 ijerph-17-07652-f001:**
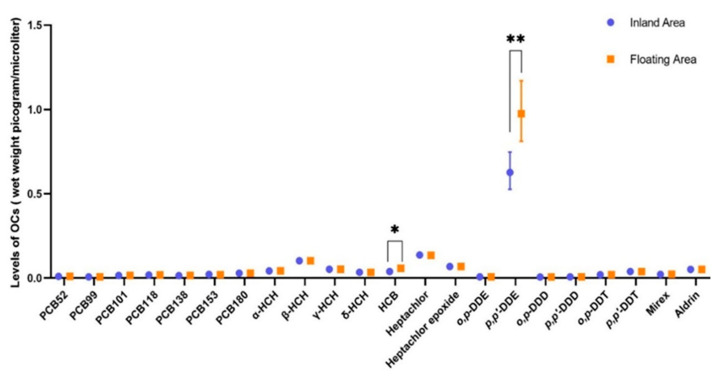
Geometric means of the organochlorine compounds concentration in mothers living in inland and floating area (wet weight picogram/microliter). The vertical bars plot the 95% confidence intervals. * *p* = 0.002; ** *p* < 0.001, there are significant differences of maternal blood hexachlorobenzene (HCB) and p,p’-DDE levels between inland and floating areas in Mann–Whitney U test.

**Table 1 ijerph-17-07652-t001:** Sociodemographic and anthropometric characteristics of studied population.

**Quantitative Variables**	**All Locations (n = 194)**			**Inland Area (n = 120)**				**Floating Area (n = 74)**				***p* Value ***
	**N (Missing)**	**Mean (SD)**	**Median (Min–Max)**	**N (Missing)**	**Mean (SD)**		**Median (Min–Max)**	**N (Missing)**	**Mean (SD)**	**Median (Min–Max)**		
Maternal age	194 (0)	26.7 (6.2)	25.0 (17.0−44.0)	120 (0)	26.7 (6.5)		25.5 (17−44)	74 (0)	26.6 (5.7)	25.0 (18–43)		0.787
Maternal weight (kg)	194 (0)	58.8 (8.1)	58.0 (41.0−84.0)	120 (0)	60.3 (7.9)		60.0 (45.0–84.0)	74 (0)	56.4 (8.0)	56.0 (41.0–79.0)		0.002
Maternal height (cm)	194 (0)	155.3 (5.2)	155.0 (145−167)	120 (0)	156.9 (4.8)		156.0 (145–167)	74 (0)	152.6 (4.7)	153.5 (145–165)		<0.001
Maternal BMI (kg/m^2^)	194 (0)	24.4 (3.0)	24.1 (17.7−34.7)	120 (0)	24.5 (2.9)		24.2 (18.3–32.8)	74 (0)	24.2(3.2)	24.1 (17.7–34.7)		0.527
Education (years)	156 (38)	6.3 (3.0)	6.0 (0−13)	108 (12)	7.1 (3.0)		7 (0–13)	48 (26)	4.6 (2.0)	5.0 (1–9)		<0.001
Gravida	194 (0)	2.5 (1.8)	2.0 (1.0−10.0)	120 (0)	2.1 (1.5)		2 (1–10)	74 (0)	3.1 (2.1)	2.0 (1–9)		0.001
Gestational age (weeks)	144 (50)	39.1 (1.0)	39.1 (38−42)	79 (41)	40.0 (0.8)		40.0 (38–42)	65 (9)	38.4 (0.7)	38.0 (38–40)		<0.001
Birth weight (kg)	149 (45)	3.1 (0.4)	3.0 (2.0−4.4)	83 (37)	3.2 (0.4)		3.1 (2.4–4.4)	66 (8)	3.0 (0.4)	3 (2.0–3.8)		0.001
Birth length (cm)	147 (47)	48.9 (2.9)	49.0 (32.0−58.0)	83 (37)	48.5 (3.4)		49.0 (32.0–54.0)	64 (10)	49.5 (2.2)	49.0 (46.0–58.0)		0.311
Head circumference	147 (47)	31.7 (2.9)	32.0 (24.7−54.0)	82 (38)	32.8 (2.6)		33.0 (28.0–54.0)	65 (9)	30.2 (2.6)	30.0 (27.0–39.0)		<0.001
Ponderal index	147 (47)	26.9 (7.3)	25.5 (15.08−91.55)	83 (37)	28.6 (8.9)		27.0 (21.6–91.55)	64 (10)	24.7 (3.4)	24.7 (15.1–33.7)		<0.001
**Categorical variables**	**Category**			**All locations (n = 194)**				**Inland area (n = 120)**			**Floating area (n = 74)**	
				**Count**		**Percentage**		**Count**	**Percentage**		**Count**	**Percentage**
Parity	Para 0/Para (1−3)/Para (≥4)			82/93/19		42.3/47.9/9.8		59/56/5	49.2/46.7/4.2		23/37/14	31.1/50.0/18.9
Newborns gender	Boy/Girl/Missing data			81/91/22		41.7/46.9/11.3		58/44/18	48.3/36.7/15.0		23/47/4	31.1/63.5/5.4
Occupation	Housewife/Workers/Farmers/Teachers/Fishing			67/29/55/4/39		34.5/14.9/28.4/2.1/20.1		34/28/54/4/0	28.3/23.3/45.0/3.3/0		33/1/1/0/39	44.6/1.4/1.4/0/52.7
Water source	River and Pond/Rainwater/Well/Bottled			77/64/45/8		39.7/32.9/23.2/4.1		8/64/45/3	6.7/53.3/37.5/2.5		69/0/0/5	93.2/0/0/6.8
Insecticide use	No/Yes			85/109		43.8/56.2		70/50	58.3/41.7		15/59	20.3/79.7
Where use the insecticide	Home and farm/Home only			36/73		33.0/67.0		35/15	70/30		1/58	1.7/98.3
Food source	In village/Market/Missing data			123/70/1		63.4/36.1/0.5		119/1/0	99.2/0.8/0		4/69/1	5.4/93.2/1.4

* Mann–Whitney U test.

**Table 2 ijerph-17-07652-t002:** Blood levels (wet weight picogram/microliter) of polychlorinated biphenyls (PCBs), hexachlorocyclohexane (HCH) and organochlorine pesticides (OCPs) in the study population (n = 194).

Contaminants	GM *	95% Confidence interval *	Median	Minimum	Maximum
PCB 52	0.010	0.010–0.010	0.010	0.010	0.010
PCB 99	0.008	0.008–0.008	0.008	0.008	0.055
PCB 101	0.016	0.016–0.016	0.016	0.016	0.020
PCB 118	0.018	0.017–0.019	0.015	0.015	0.112
PCB 138	0.015	0.014–0.017	0.012	0.012	0.601
PCB 153	0.021	0.019–0.023	0.014	0.014	1.173
PCB 180	0.029	0.028–0.030	0.028	0.028	0.423
α-HCH	0.043	0.043–0.043	0.043	0.043	0.043
β-HCH	0.103	0.103–0.103	0.103	0.103	0.103
γ-HCH	0.053	0.053–0.053	0.053	0.053	0.053
δ-HCH	0.034	0.034–0.035	0.034	0.034	0.070
HCB	0.046	0.042–0.049	0.044	0.010	0.196
Heptachlor	0.136	0.135- 0.138	0.135	0.135	0.352
Heptachlor epoxide	0.069	0.069–0.069	0.069	0.069	0.069
*o,p*-DDE	0.008	0.008–0.008	0.008	0.008	0.010
*p,p’*-DDE	0.742	0.651– 0.846	0.720	0.020	8.047
*o,p*-DDD	0.007	0.007–0.008	0.007	0.007	0.018
*p,p’*–DDD	0.008	0.008–0.008	0.008	0.008	0.010
*o,p*-DDT	0.020	0.020–0.020	0.020	0.020	0.020
*p,p’*-DDT	0.038	0.036–0.041	0.033	0.030	0.519
Mirex	0.023	0.022–0.023	0.022	0.020	0.113
Aldrin	0.052	0.052–0.052	0.052	0.050	0.052

* Geometric mean with 95% confidence intervals.

**Table 3 ijerph-17-07652-t003:** Multiple linear regression models showing the associations of sociodemographic and other maternal variables with maternal organochlorine (OC) levels.

	PCB 99 ^a^	PCB 101 ^a^	PCB 118 ^a^	PCB 138 ^a^	PCB 153 ^a^	PCB 180 ^a^	HCB ^a^	Heptachlor ^a^	*o,p*-DDE ^a^	*p,p’*-DDE ^a^	*o,p*-DDD ^a^	*p,p’*-DDD ^a^	*p,p*’-DDT ^a^	Mirex ^a^	Aldrin ^a^
**Age**	0.10	−0.01	−0.07	0.17	**0.26 ***	0.19	0.15	0.01	0.06	**0.39 ****	0.18	0.22	0.22	−0.22	−0.22
**BMI**	−0.06	−0.01	0.04	−0.03	−0.06	−0.08	0.14	−0.07	−0.17	0.07	−0.05	−0.22	−0.11	0.07	0.22
**Parity ^b^**															
Para (1−3)	−0.13	0.08	−0.07	**−0.24 ***	**−0.33 ***	−0.17	**−0.29 ***	−0.08	0.17	**−0.45 ****	−0.05	0.05	0.002	0.10	−0.05
Para (≥4)	−0.08	−0.02	−0.02	−0.20	**−0.31 ***	−0.12	**−0.25 ***	−0.02	0.03	**−0.45 ****	−0.08	−0.04	−0.11	**0.42 ****	0.04
**Education**	0.12	−0.16	−0.04	0.07	0.07	0.10	0.07	0.08	−0.09	**0.21 ***	−0.14	−0.01	**0.24 ***	−0.04	0.01
**Residence area ^c^**	0.07	−0.02	−0.23	−0.28	−0.24	0.06	0.33	0.13	−0.04	−0.08	−0.17	−0.03	−0.16	0.05	0.03
**Insecticide use ^d^**	0.05	0.07	0.11	0.06	0.002	0.13	−0.08	−0.09	−0.18	**0.19 ***	−0.17	−0.12	0.12	0.03	0.12
**Food source ^e^**	0.04	−0.03	−0.01	**0.52 ***	0.36	0.04	0.35	−0.02	0.05	0.34	0.11	0.09	0.14	−0.29	−0.09
**Water source ^f^**															
Rainwater	−0.13	0.08	−0.01	−0.10	−0.09	−0.12	−0.14	−0.05	0.06	0.15	−0.28	−0.04	−0.12	0.06	0.04
Well	−0.10	0.18	−0.09	−0.09	−0.13	−0.09	−0.02	0.03	0.07	0.08	−0.16	−0.09	−0.01	−0.01	0.09
Bottled	−0.05	0.03	−0.00	−0.04	0.07	−0.05	−0.04	−0.01	0.003	0.04	−0.13	−0.04	−0.13	−0.03	0.04
**Occupation ^g^**															
Worker	0.02	−0.01	−0.11	0.00	0.15	0.03	0.12	0.13	−0.01	0.12	−0.27	0.02	−0.14	−0.04	−0.02
Farmers	0.14	0.18	0.02	0.12	0.06	0.09	−0.10	0.08	0.23	−0.08	−0.27	0.14	−0.20	−0.05	−0.14
Teacher	−0.04	0.07	**0.22 ***	0.07	**0.19 ***	−0.04	0.07	−0.02	0.03	0.01	−0.09	−0.01	0.05	0.03	0.01
Fishing	0.01	−0.01	0.07	−0.11	−0.06	0.003	−0.12	0.002	−0.01	0.01	−0.04	−0.01	0.05	**0.24 ***	0.01

Values shown were standard multiple regression analyses coefficient. ^a^ Persistent organic pollutants (POPs) were log 10 transformed before analysis. ^b^ Para 0 as the reference category. ^c^ Inland as reference category for residence. ^d^ Participants who do not use insecticide as reference. ^e^ Food source from village as reference. ^f^ Using river or pond water as reference. ^g^ Women who work as housewife as reference. * *p* < 0.05; ** *p* < 0.001. Significant findings highlighted in bold.

**Table 4 ijerph-17-07652-t004:** Multivariable linear analysis of selected POPs and pregnancy outcomes; adjusted for maternal age, parity, BMI, residential area, education and occupation.

	Gestational Age ^b^ (n = 108)		Birth Weight ^c^ (n = 109)		Birth Length ^c^ (n = 108)		Head Circumference ^c^ (n = 108)		Ponderal Index ^c^ (n = 108)	
	Std. β ^d^ (95% CI)	*p* Value	Std. β^d^ (95% CI)	*p* Value	Std. β^d^ (95% CI)	*p* Value	Std. β^d^ (95% CI)	*p* Value	Std. β ^d^ (95% CI)	*p* Value
**PCB 99 ^a^**	**−0.16 (−2.51 to −0.07)**	**0.039**	−0.01 (−0.68 to 0.62)	0.930	−0.14 (−9.13 to 1.68)	0.174	−0.03 (−5.61 to 3.97)	0.735	0.09 (−8.41 to 20.69)	0.404
**PCB 101 ^a^**	0.06 (−10.27 to 22.64)	0.457	0.08 (−4.69 to 12.02)	0.386	0.01 (−66.65 to 75.11)	0.906	0.02 (−54.08 to 70.37)	0.796	−0.01 (−201.05 to 178.20)	0.905
**PCB 118 ^a^**	−0.02 (−1.15 to 0.94)	0.842	−0.08 (−0.71 to 0.33)	0.468	−0.22 (−8.79 to −0.19)	**0.041**	0.06 (−2.73 to 4.98)	0.564	0.21 (−0.35 to 22.73)	0.057
**PCB 138 ^a^**	−0.24 (−1.28 to −0.32)	**0.003**	−0.04 (−0.38 to 0.25)	0.680	−0.16 (−4.70 to 0.66)	0.139	0.002 (−2.36 to 2.40)	0.986	0.10 (−3.98 to 10.46)	0.375
**PCB 153 ^a^**	−0.18 (−1.06 to −0.05)	**0.032**	0.02 (−0.24 to 0.29)	0.856	−0.20 (−4.31 to 0.07)	0.058	−0.092 (−2.90 to 1.02)	0.343	0.20 (−0.42 to 11.30)	0.068
**PCB 180 ^a^**	−0.15 (−3.16 to 0.02)	0.052	−0.07 (−1.15 to 0.51)	0.446	−0.20 (−14.07 to −0.35)	**0.040**	−0.02 (−7.01 to 5.31)	0.785	0.12 (−7.50 to 29.77)	0.239
**HCB ^a^**	0.06 (−0.47 to 0.90)	0.527	0.001 (−0.35 to 0.35)	0.990	−0.02 (−3.15 to 2.74)	0.889	−0.07 (−3.53 to 1.63)	0.466	0.01 (−7.42 to 8.33)	0.908
**Heptachlor ^a^**	−0.09 (−5.42 to 1.49)	0.262	−0.01 (−1.86 to 1.69)	0.928	−0.02 (−16.55 to 13.41)	0.836	0.02 (−11.70 to 14.64)	0.825	0.02 (−36.58 to 43.59)	0.863
**o,p-DDE ^a^**	−0.21 (−33.19 to −4.43)	**0.011**	0.08 (−4.27 to 10.35)	0.411	−0.32 (−156.86 to −39.49)	**0.001**	0.04 (−42.28 to 66.47)	0.660	0.35 (121.95−433.79)	**0.001**
**p,p’-DDE^a^**	0.02 (−0.30 to 0.40)	0.776	−0.06 (−0.23 to 0.12)	0.548	0.06 (−1.05 to 1.94)	0.558	0.03 (−1.10 to 1.52)	0.750	−0.08 (−5.43 to 2.55)	0.475
**o,p-DDD ^a^**	−0.15 (−6.73 to 0.28)	0.071	0.21 (0.08−3.64)	**0.041**	−0.08 (−21.37 to 9.28)	0.436	−0.004 (−13.75 to 13.24)	0.970	0.15 (−11.02 to 70.35)	0.151
***p,p’*-DDD ^a^**	−0.10 (−23.42 to 4.96)	0.200	0.04 (−5.58 to 8.46)	0.684	−0.25 (−132.95 to 18.26)	**0.010**	0.06 (−33.59 to 70.40)	0.484	0.28 (64.81−369.92)	**0.006**
**p,p’-DDT ^a^**	−0.11 (−1.71 to 0.25)	0.144	−0.02 (−0.56 to 0.46)	0.849	−0.07 (−5.87 to 2.68)	0.462	−0.10 (−5.87 to 1.62)	0.262	0.04 (−9.46 to 13.47)	0.729
**Mirex ^a^**	−0.05 (5.00 to 2.65)	0.544	0.06 (−1.34 to 2.54)	0.542	0.03 (−13.99 to 18.91)	0.767	−0.09 (−21.36 to 7.40)	0.338	0.003 (−43.48 to 44.57)	0.980
**Aldrin ^a^**	0.10 (−32.18 to 151.99)	0.200	−0.04 (−54.93 to 36.21)	0.684	0.25 (118.52–862.91)	**0.010**	−0.06 (−456.9 to 218.03)	0.484	−0.28 (−2400.97 to −420.65)	**0.006**

^a^ POPs were log10 transformed before analysis. ^b^ Gestational age model adjusted for maternal age, parity, BMI, residence area, education, occupation, and ponderal index. ^c^ Birth weight, length, head circumference, and ponderal index models adjusted for gestational age, maternal age, parity, BMI, residence area, education, and occupation. ^d^ β coefficients of the multiple regression models after standardizing all the variables. Significant findings highlighted in bold.

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
