# Peer review of "Serum Concentrations of Selected Organochlorines in Pregnant Women and Associations with Pregnancy Outcomes. A Cross-Sectional Study from Two Rural Settings in Cambodia"

_ijerph, 2020, doi:10.3390/ijerph17207652_

Round 1
Reviewer 1 Report
Title:
Line 1-4: Serum concentrations of selected organochlorines in 2 pregnant women and associations with pregnancy 3 outcomes. A cross-sectional study from two rural 4 settings in Cambodia.
Comments: Reverse line 1-4 as
“The concentrations of selected persistent organic pollutants in the serum of pregnant women and their effects on pregnancy 3 outcomes. A cross-sectional study from two rural 4 settings in Cambodia”
Or
“The concentrations of organochlorines pesticides and polychlorinated biphenyls in the serum of pregnant women and their health implications on pregnancy 3 outcomes. A cross-sectional study from two rural 4 settings in Cambodia
Abstract
Line 19-31
Comment
From here
Serum samples (194) were collected from pregnant women in the low-income North and Western areas in Cambodia. The samples were extracted using solid phase extraction technique and cleaned-up. The six organochlorine pesticides (OCPs) and seven polychlorinated biphenyl (PCBs) congeners in the extract were analyzed with a Shimadzu GC 2010 gas chromatography mass spectrometry (GC-MS). It was coupled with SLB-5 MS 102 column (30m 0.25 m0.25 μm) operated in selected ion monitoring (SIM) mode and positive electron-impact ionization (EI+). The concentrations of OCPs and PCBs detected were in the range from 0.1 to 0.2 mg/l and from 0.4-5 mg/l respectively. In maternal serum, the results revealed low level of PCBs and some OCPs analytes except for heptachlors, β-hexachlorocyclohexane (HCH), heptachlor epoxide and p,p’-DDE. ………..complete with your results
Introduction
Line 51-57
Both synthetic substances like organochlorines (OCs) and naturally occurring non-essential toxic trace elements are classified as PTS due to their properties; persistence, their half-lives are very long under most environmental conditions; bioaccumulation, increasing levels of the substance as the individual grows older; biomagnification, the concentration increases in the food chain; toxic, the substance may be toxic in very small amounts. The long-term impact of exposure to these pollutants is not fully understood. Of particular interest is the “cocktail effect” on the developing neural systems in fetuses and young children since organochlorines are lipophilic and thus accumulates in fatty tissue.
Line 51-57: Rephrase
For instance, persistent organic pollutants (POPs) stay for a long time in our environment due to their half-lives and bioaccumulate in food chain because their fat soluble.
Line 60: In this article we will look at levels of selected organochlorines in the blood of pregnant women.
Comment: Rephrase as ………………………….“selected POPs (OCPs and PCBs) in the blood…………….
Line 66 -70 : Cambodia is a small low-income country in South East Asia with a population of 16,2 million people. It is ranked as number 146 out of 189 countries in human health indicators. According to United Nations Development Program (UNDP), close to 40% of Cambodia’s population live in so called multidimensional poverty, and 32.4 % of all children < 5 years of age suffer from moderate or severe malnutrition or stunting [9, 10].
Comment: Move line 66-70 to start the paragraph line 59.
Cambodia is a small low-income country in South East Asia with a population of 16,2 million people. It is ranked as number 146 out of 189 countries in human health indicators. According to United Nations Development Program (UNDP), close to 40% of Cambodia’s population live in so called multidimensional poverty, and 32.4 % of all children < 5 years of age suffer from moderate or severe malnutrition or stunting [9, 10]. Vulnerable populations are more exposed to pollutants both ambient and through the food chain. In this article we will look at levels of selected organochlorines in the blood of pregnant women from two rural communities in Cambodia and the effect on birth outcomes as gestational age, length, weight, ponderal index and head circumference of their babies. We will also discuss possible sources of exposure from environmental surroundings and diet.
Line 81 : The data collection took place at the local health centers in the villages. A total of 194 women,
Comment: Rephrase as “Samples were collected from local health centers in the villages”
Line 82: A total of 194 women, 120 from area one and 74 from area two, were recruited while in the third trimester of the pregnancy.
Comment: Rephrase
Line 80: into two separate sterile BD vacutainer® plastic tubes; the EDTA tube for plasma and another tube.
Comment: Rephrase to reflect why EDTA tube is used for plasma and another “named” tube
Line 100-105: GC-MS
Comment: Rephrase to capture, oven program, injection volume and temperature as well as quality control procedure carried out to achieve excellent results.
Line 121-123: Complete case analysis was used for handling missing data, which means that participants with any missing data were excluded in the statistical analyses.
Comment: Rephrase
Line 125 -134
Ethical approval
The participant’s consent was given both orally and in writing. The informed consent forms 126
were presented to every woman by local health staff to ensure full understanding of the study aims and process. The document was signed or thump printed on site when the blood samples were drawn. The consent forms and written files of demographical and laboratory data were stored in locked room at the Trauma Care Foundation head office in Battambang, Cambodia. Access to non nanonymous data was restricted to members of the research team. The data was stored and processed according to the approval in National Ethics Committee for Health Research of the Ministry of Health, Cambodia (ref. 0365 N.E.C.H.R., 29/12/2014 and 114 N.E.C.H.R; 28/03/2016), as well as the Regional Committee for Medical Research Ethics, 2016, ref: 2015/2486/ REK Nord, Norway.
Comments: the ethical approval should come be collection of line 80 Data collection/ Sample collection, rearrange please.
Line 165: ….. POPs…..
Comment: Replace as selected POPs
Line 167: The most abundant POPs
Comment: Replace as ”The most abundant OCPs”
Line 172: Table 2:
Comment: Replace concentrations of “POPs” with “PCBs and OCPs”
Line 174: ……………..The sentence
Comment: Rephrase.
Line 218-219: ……. The sentence
Comment: Revisit/rephrase
Line 249: significant associations were found for the studied POPs and the babies’ head circumference.
Comment: Specify the studied analytes OCPs/PCBs. POPs is an ambiguous
Discussion
Line 251-252: In our study we found low levels of polychlorinated biphenyls (PCB) and organochlorine pesticides (OCP), except for heptachlors, β-HCH, heptachlor epoxide and p,p’-DDE
Comment: Rephrase / revise as “ In our study we found low levels of polychlorinated biphenyls (PCB) and o,p’-DDE and p,p’-DDD, mirex and aldrin (least them pls) except for heptachlors, β-HCH, heptachlor epoxide and p,p’-DDE.
Line 251-349: Kindly revisit.
Yes, you cite relevant paper but you the not fix them at the end of your finding. Place them in appropriately not at the of your findings. Download Yahaya et al., 2017.
Yahaya, A., Okoh, O.O.., Okoh, A.I., Adeniji .O.A. Occurrences of Organochlorine pesticides along the course of Buffalo River in the Eastern Cape of South Africa and its health implications. Int. J. Environ. Res. Public Health,
2017, 14, 1372: 1-16. doi:103390/ijerph14111372.
Conclusion
Line 350-359: Revise it.
Comment: Avoid using POP anyhow, it’s confusing. Know when to use selected POPs or OCPs and PCBs. Your conclusion is scanty and note your method of extraction was not captured anywhere either SPE or Liquid-liquid technique.
Best of luck
Author Response
Dear colleague,
Please find the attached document where we have replied to your review
Reviewer 2 Report
- Abstract: use of many jargons and names of different chemical components should be avoided. These might not be understandable to many readers. Meaning of p,p′-DDE, p,p′-DDD, o,p′-DDE, o,p′-DDD were not realized.
- Introduction: Page 2, Lines 51-55: long sentence, difficult to understand the meaning. Needs to break down in small, meaningful sentences.
- Methods:
- Study site: Page 2, Lines 66-67, population and people are repetitive. Rewrite the sentence
- Page 2, Line 71, duration October 2015 to 2017 (? October). Be specific.
- What was the sampling criteria?
- What is EDTA?
- Blood sample was collected at the time of inclusion, i.e. since October 2015? And the samples were transferred on 28 October 2016. Where were these samples stored during the interim period? The data collection process and steps are not clear.
- Full form of POP, not written anywhere
- Page 3, Line 128, Thumb, not thump.
- Overall Methods part needs re-writing. Not clear.
- Results:
- Page 6, Line 159, there were 172 newborns included in the study. What about the other 22 pregnant women? Because total sample was 194.
- Discussion:
- Page 14, Line 343, correct the spelling of development.
- Recommendations and conclusions should be written with more assertive language to be able to influence policy. Some recommendations on the effects of insecticide use should come.
- There are many jargons, and generic names throughout the paper, those should be avoided.
Author Response
Dear colleague,
Please find the attached document where we have replied to your review!

Reviewer 3 Report
In the present study, the Authors conducted a cross-sectional study among 194 pregnant women from Chroy Sdao district (120 women) and "the floating villages" (74 women) in order to study organochlorines levels in maternal serum and correlate with birth outcomes. They described that women from the floating areas had significantly higher blood concentrations of p,p’-DDE and HCB compared to women situated in the inland villages and that parity, age, residential area and educational level are direcly correlated with pollutants exposure. Moreover, the Authors found a significant negative association between the PCB congeners 99, 138 and 153 and gestational age and between gestational age, birth length and maternal blood levels of o,p’-DDE. Finally, They demonstated a positive associations between o,p’-DDD and birth weight and that both p,p’-DDD and o,p’-DDE were positively associated with the baby’s ponderal index. They concluded that women from the poorest settings have higher blood concentrations of pollutants and this is linked to negative pregnancy outcomes.
This is a very interesting study addressing a novel issue as organochlorines effects during human pregnancies. Thus, it is likely to be of great interest to the readers of IJERPH.
However, there are several points that the Authors must address before publication.
- The main concern is about the study population. 194 pregnant women is not a small study population but there is a great disparity in the women belonged to the group a (120) and b (74). It could have altered statistical outputs. Please, clarify
- The Authors aimed to correlate organochlorines exposure to pregnancies and fetal outcomes however you did not mentioned if all the women enrolled were physiological or developed some pregnancies pathologies such as gestational diabase, preeclampsia. Please, clarify.
- The Authors stated that collected blood and urine samples in the third trimester of pregnancies for all women enrolled as confimed by the mean and median of gestational age that are quite similar. However They reported a significant difference in gestational age (p<0.001) between group 1 and 2 in Table 1. Please, clarify.
- It is well known that prenatal exposure to organochlorines occurs during pregnancy resulted in transfer of these chemicals to the embryo and fetus through the placenta (https://www.ncbi.nlm.nih.gov/pmc/articles/PMC4097106/pdf/nihms583119.pdf). Data for placental weight are available?
- In the abstract section you should specify in which group you find differences (lines 24-25) (line 26)(line 29).
- Please, check if birth weight is really statistically different between group 1 (mean:3.2) and group 2 (mean: 3).
- In the discussion section, the Authors only mentioned other study that are in agree or disagree with their results but nothing, except parity (lines 315-320), is commented.
- English must be revised.
Author Response

(The authors gave the same response as above.)

Reviewer 4 Report
Thank you very much for giving me the opportunity to review the manuscript entitled “Serum concentrations of selected organochlorines in pregnant women and associations with pregnancy outcomes. A cross-sectional study from two rural settings in Cambodia”
Please see below my specific comment:
- In the abstract some statistics for observed associations need to be added (p value or 95% CI)
- The information about the time period for exposure assessment needs to be added into the abstract of the manuscript
- In my opinion the first part of the conclusion in the abstract is not really clear (“The most vulnerable population had higher exposure”) – this need to be modified. I suppose the authors want to point out that poorest areas had significantly higher exposure.
- The sentences in lines 48-49 are not related to the study. In that short introduction the authors should rather focus on birth outcomes not on neurotoxicity
- I would change “In this article..” into “In this study..”(line 60)
- The authors used the term cross-sectional study (which by definition is the study where exposure and outcome is evaluated at the same time point) – in the current study exposure was evaluated in 3rd trimester of pregnancy and then birth outcomes were evaluated – so it that case it is rather cohort design. What was the rationale for using the term cross-sectional?
- The methods part of the manuscript needs to be modified. There is no information about questionnaire data collected as well as outcome measures (how they are defined). I would propose to add additional subheadings: exposure assessment (so you can move relevant information here), outcome assessment, sociodemographic variables (or covariate).
- The covariates and confounders needs to be listed in statistical analysis section of the manuscript.
- The first sentence under the results (lines 139-140) is rather repetition of that what is presented in methods part (so this sentence can be deleted).
- The last part of the discussion needs to be rewritten – I would rather clearly point out strengths and limitations of the study.
Author Response

(The authors gave the same response as above.)

Round 2
Reviewer 2 Report
The authors have improved the manuscript according to my suggestions. So the paper can be accepted in its current form.
Author Response
Thanks a lot for your valuable contribution!
Reviewer 3 Report
The Reviewer would like to thank the authors for addressing all the comments. Although the authors did not check if birth weight is really statistically different between group 1 (mean:3.2) and group 2 (mean: 3). Please, clarify.
Author Response
Dear colleague,
We have run the statistics over again, and we have added a clarifying sentence in the manuscript (line 249.....) We hope this is sufficient?
. Although arithmetic average birth weight was similar in inland and floating area (3.2 kg vs. 3.0kg), neonatal birth weight from inland area has a larger mean rank (85.1) than those in floating area with mean rank (62.3) (Data not shown), a statistically significant different was observed (U =1901.0 , p = 0.001).
Reviewer 4 Report
The authors have improved the manuscript according to my suggestions.
In my opinion paper can be accepted in current form.
Author Response
Thanks a lot for your valuable contributions!